# Availability of results of interventional studies assessing colorectal cancer from 2013 to 2020

Anna Pellat[1,2]*, Isabelle Boutron[2,3], Philippe Ravaud[2,3]

**1** Gastroenterology and Digestive Oncology Unit, Assistance Publique des Hôpitaux de Paris, Cochin Teaching Hospital, Université de Paris, Paris, France, **2** Centre of Research in Epidemiology and Statistics (CRESS), Inserm U1153, Université de Paris, Paris, France, **3** Centre d'Épidémiologie Clinique, Assistance Publique des Hôpitaux de Paris, Hôtel Dieu Hospital, Université de Paris, Paris, France

\* anna.pellat@gmail.com

## Abstract

Colorectal cancer (CRC) is one of the most frequent cancers worldwide. Our aim was to evaluate the availability of results of interventional studies studying CRC. We searched the ClinicalTrials.gov registry for all interventional studies on CRC management in adults completed or terminated between 01/01/2013 and 01/01/2020. To identify results, we searched for results posted on the ClinicalTrials.gov registry and/or published in a full-text article. Our primary outcome was the proportion of CRC interventional studies with available results (i.e. posted on the ClinicalTrials.gov registry and/or published in a full-text article). Secondary outcomes were 1) median time between primary completion and earliest date of results availability, 2) the cumulative percentage of interventional studies with results available over time 3) the cumulative percentage of interventional studies with results posted on the ClinicalTrials.gov registry over time and 4) the percentage of results available in open access. We identified 763 eligible interventional studies in ClinicalTrials.gov, which included 679 198 patients. Of these, 286 (37%) trials, including 270 845 (40%) patients, did not have any results available. Median time for results availability was 32.6 months (IQ 16.1-unreached). The cumulative percentage of interventional studies with available results was 17% at 12 months, 39% at 24 months and 55% at 36 months. Results were more likely available for trials that were randomized, completed, had one trial site in the United States, and with mixed funding. The cumulative percentage of interventional studies with results posted on Clinical-Trials.gov was 2% at 12 months. Results were available in open access for 420 (420/477 = 88%) trials. Our results highlight an important waste in research for interventional studies studying CRC.

**Data Availability Statement:** We have uploaded out dataset on the Zenodo repository with the following link https://doi.org/10.5281/zenodo.5883227.

## Background

Since the end of the 20th century, cancer has become an important public health issue worldwide. Colorectal cancer (CRC) is one of the most frequent malignant neoplasms with varying incidences and mortality rates across countries. Overall, it is the third most commonly diagnosed cancer in males and the second in females with more than 1.9 million new cases in 2020

**Funding:** The authors received no specific funding for this work.

**Competing interests:** The authors have declared that no competing interests exist.

according to the World Health Organization Global Cancer Observatory (GCO) database (*https://gco.iarc.fr/*). Organized screening as well as the development of various treatments (surgery, chemotherapy targeted therapies and immunotherapy) have improved prognosis and quality of life of CRC patients, but approximately 935 000 deaths were still registered in 2020.

Since the 2004 announcement by the International Committee of Medical Journal Editors of required registration for each new trial before participant enrollment and as a condition for publication, there has been a strong increase of new trial registrations worldwide with a better identification of the ongoing research [1–3]. The ClinicalTrials.gov registry, developed by the United states (US) National Institutes of Health (NIH), is currently the largest used registry [1, 3]. In the oncology field, there has been an increased number of conducted interventional studies over time, including a growing number of pilot trials and industry-sponsored trials [4–6]. As a consequence, cancer drugs have comprised the single largest category of new drug approvals in Europe in the recent years [6].

However, previous works have highlighted an important waste in the production and reporting of research in various fields because of lack of quality and standardization in the different research steps [7–9]. Among others, this waste could be related to inaccessible research results with a risk of biased evidence and literature [7, 9–11]. Trial results can be accessible through registries and/or published in various supports (abstracts, full-text articles, summary reports...). Results availability has never been evaluated for CRC interventional studies. The aim of our work was to assess the availability of results for interventional CRC studies registered on the ClinicalTrials.gov registry and completed or terminated between 01/01/2013 and 01/01/2020.

## Methods

### Search strategy and sample identification

We searched the United States (US) National Library of Medicine database of clinical trials, ClinicalTrials.gov, on the 11[th] of March 2021, for all interventional studies completed or terminated between 01/01/2013 and 01/01/2020 and studying CRC management in adults (S1 Table).

### Data extraction

With our previous search, we downloaded the list of encountered interventional studies, all referenced on ClinicalTrials.gov with a unique identification code or NCT number. In order to create a personalized database, we used the aggregate analysis of ClinicalTrials.gov (AACT) database from the Clinical Trials Transformation Initiative (CTTI), which is a publicly available relational database containing all information about every study registered in ClinicalTrials.gov. Using the Beaver software (SQL language), we extracted selected items from the AACT database for the previous list of encountered interventional studies, based on their NCT number. The selected extracted items are listed in our Data extraction form (S2 Table).

### Eligibility criteria

Inclusion criteria for our sample were: all completed or terminated interventional studies, performed in adults (16 years old and over), of any phase, mentioning the type of patient enrollment (actual or estimated), focusing on CRC management, and with a primary completion date between 01/01/2013 and 01/01/2020. We defined as management all trials with a primary purpose of screening, diagnosis, prevention (secondary and tertiary), treatment (including drugs,

dietary supplement, behavioral intervention, device and surgery), and supportive care. Of note, these are response options proposed by ClinialTrials.gov for primary purpose and treatment/intervention types. When the primary purpose of the study was not specified ("other" or not available), we checked that each trial's primary purpose fitted our previous definition.

As stated on ClinicalTrials.gov on the 11[th] of March 2021, primary completion date was defined as the date on which the last participant in the study was examined or received an intervention to collect final data for the primary outcome measure. Whether the clinical study ended according to the protocol or was terminated did not affect this date. For interventional studies with more than one primary outcome measure with different completion dates, this term refers to the date on which data collection is completed for all the primary outcome measures. The estimated enrollment was defined as the target number of participants that the researchers needed for the study.

We then excluded interventional studies focusing on different conditions (e.g. mixed malignancies, surgery of benign colorectal lesions, adenoma detection outside of CRC screening or cancer predisposition syndromes). The identification of eligible interventional studies was done by one independent reviewer and checked with a senor reviewer.

## Identification of results availability

To identify results in our sample, we systematically searched for results posted on the ClinicalTrials.gov registry and/or published online in a full-text article.

We first identified all interventional studies with posted results on the Clinicaltrials.gov registry (information extracted from the AACT database).

We then searched for online publication of results for the whole sample of eligible interventional studies, even for those with posted results on ClinicalTrials.gov, in order to identify the earliest date of results availability. If more than one publication of results was identified for one trial, we kept the earliest date of publication.

To identify publication of results, we used the publication link in ClinicalTrials.gov when available (direct access to PubMed publication). If a publication of results was found, the search was stopped. If no link was available on ClinicalTrials.gov, we also systematically searched MEDLINE via PubMed and Google Scholar using keywords for treatment and/or drug names, the principal investigator's last name and the condition studied. If necessary, for industry-sponsored trials, we also searched the sponsor's website via Google to look for the final results of industry-funded trials. We also used keywords for treatment and/or drug names and the condition studied on the sponsor's website. All interventional studies without results in a full-text article were censored on April 15[th], 2021.

**Eligibility criteria for results.**   Regarding results posted on the ClinicalTrials.gov registry, we only considered fully accessible final results (any type of results). We did not include submitted results undergoing quality control check by ClinicalTials.gov.

Regarding online publication of results outside of the registry, we considered publications reporting any type of results for the interventional study. All online publications identified were assessed by one reviewer who determined if 1) the corresponding interventional study matched in terms of the information registered (i.e., same NCT when mentioned, similar title, same studied condition, same interventions, same population, same study location, same sponsors and collaborators, same authors, and same time period) and 2) reported results published in a full-text article. We considered results to be available in open access when available on ClinicalTrials.gov and/or in an open access publication downloaded from our institution.

All doubtful cases (partial matching) were assessed by a second reviewer.

## Trial characteristics evaluated for association with results availability

To examine the association of trial characteristics with availability of results and posting of results on ClinicalTrials.gov we *a priori* selected two explanatory variables based on their interest following previous findings in the literature: type of funding and trial design. Type of funding (sponsors and collaborators) was divided in three categories: industry-funded, non-industry funded (e.g. academic or public funding) and mixed funding (both industry and non-industry funding). Design was divided in two categories: randomized versus non-randomized. We also performed a *post-hoc* analysis to evaluate the effect of three additional trial characteristics: trial location (at least one trial site in the US versus no trial site in the US), trial status (terminated or completed) and trial start date. For evaluation of the trial start date, we compared recent trials with a start date on or after 2015-01-01 with older trials (before 2015).

## Outcome measures

**Primary outcome measure.**   Our primary outcome measure was the proportion of CRC interventional studies with available results (i.e. posted on the ClinicalTrials.gov registry and/or published online in a full-text article).

**Secondary outcome measures.**   The median time between primary completion date and earliest date of results availability (i.e. posted on the ClinicalTrials.gov registry and/or published online in a full-text article).

The cumulative percentage of interventional studies with available results (posted on the ClinicalTrials.gov registry and/or published results in a full-text article) over time after the primary completion date: overall and stratified according to design (randomized versus non-randomized) and study funding (industry, non-industry and mixed).

The cumulative percentage of interventional studies with posted results on the ClinicalTrials.gov registry over time (irrespective of publication in a full-text article): overall and stratified according to design (randomized versus non-randomized) and study funding (industry, non-industry and mixed).

The percentage of results available in open access (on ClinicalTrials.gov and/or in an open access publication downloaded from our institution).

## Statistical analysis

We used the R software (R studio Version 1.2.5033) for all statistical analysis. Continuous data were given in mean with standard deviation (SD), or median with interquartile (IQ) range. Binary and categorical data were given in percentages.

For the assessment of cumulative percentages and time to event analysis we used the Kaplan Meier method. If the date of publication of results was anterior to the primary completion date (negative time for survival), we considered that results were published on the date of trial completion.

We estimated the hazard ratios (HR) by using the Cox Proportional-Hazards model to evaluate the effect of two pre-specified trial characteristics on time to event (type of funding and trial design). We also performed a *post-hoc* multivariate analysis to evaluate three additional trial characteristics: trial location, trial status and trial start date. We tested the proportional hazard assumption for each variable in our model by using the cox.zph function on R and plotting the scaled Schoenfeld residuals against time.

Because the data set included all studies in the population of interest, analytical methods were descriptive. In accordance with guidance provided by CTTI, statistical inference was not performed. Therefore, 95% confidence intervals (CI) contained in this paper represent population parameters rather than sample statistics.

## Results

### Sample identification

Our search on ClinicalTrials.gov found 1 141 interventional studies on CRC of which 763 were eligible (Fig 1).

### General characteristics of interventional studies

Among the 763 eligible interventional studies which included 679 198 patients, 621 (81%) were completed and 496 (65%) had a treatment purpose (Table 1). There were 405 (53%) randomized trials. Most trials used a parallel design (403, 53%) or single-arm design (321, 42%) and were open-labelled (585, 77%). The median sample size of enrollment was 59 (IQ 24–192), with 734 (96%) trials mentioning the actual enrollment number. A total of 512 (67%) trials were declared funded by non-industry sources. Finally, 408 (53%) trials were prospectively registered (registered before the study start date).

### Outcome results

**Primary outcome measure.** Among the 763 eligible interventional studies, 477 (63%) had results available. There were 308 (308/477 = 65%) trials with results solely published in a full-text article, 106 (22%) were posted and published and 63 (13%) were only posted on the CinicalTrials. gov registry (Fig 1). The number of patients included in the 286 trials (37%) with unavailable results was 270 845 (40%), with 270 (94%) trials mentioning the actual number of enrollments.

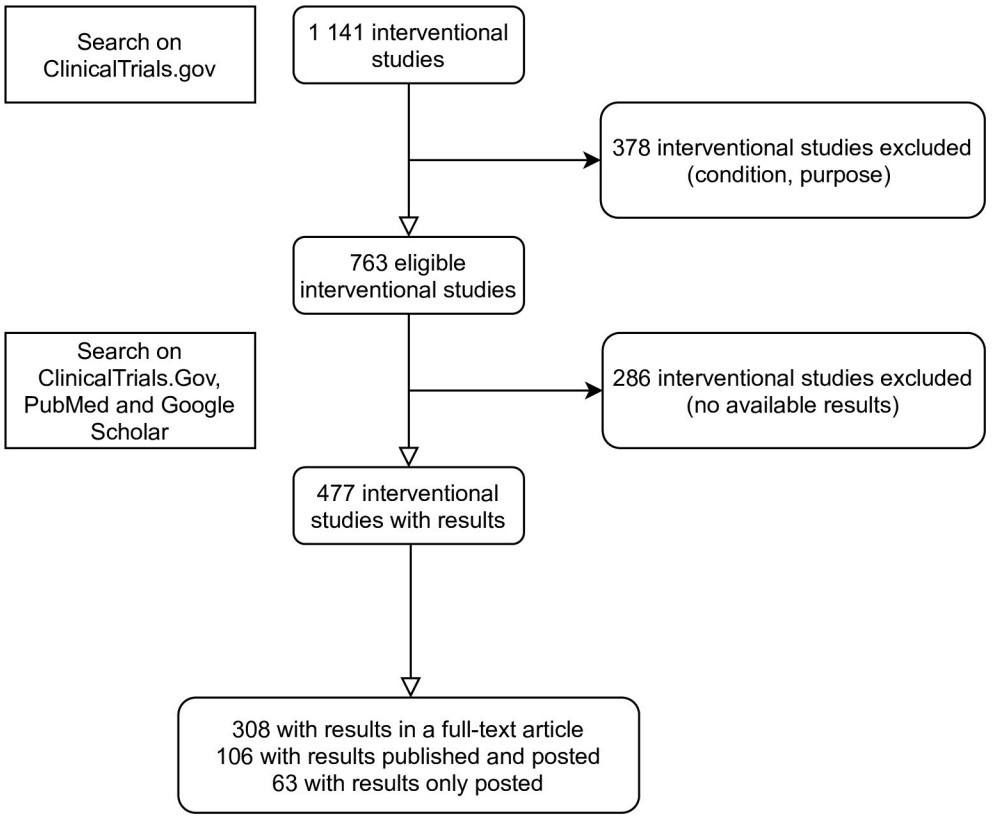

**Fig 1. Flow chart of our search.**

**Table 1. General characteristics of the eligible interventional studies (N = 763).**

| Characteristics of interventional studies | | N, (%) |
|---|---|---|
| Status | Terminated | 142 (19) |
| | Completed | 621 (81) |
| Study allocation | Randomized | 405 (53) |
| | Non-randomized | 358 (47) |
| Study design | Single group | 321 (42) |
| | Parallel | 403 (53) |
| | Cross over | 19 (2) |
| | Other[a] or NA | 20 (3) |
| Blinding | Yes (any type) | 173 (22) |
| | None (open label) | 585 (77) |
| | NA | 5 (1) |
| Primary purpose | Screening | 69 (9) |
| | Prevention | 56 (7) |
| | Diagnostic | 53 (7) |
| | Treatment | 505 (66) |
| | Supportive care | 47 (6) |
| | Other or NA[b] | 33 (5) |
| Enrollment type | Actual | 734 (96) |
| | Estimated or NA | 28 (4) |
| Trial location | At least one site in the US | 284 (37) |
| | No site in the US | 479 (63) |
| Type of funding | Industry | 127 (17) |
| | Non-industry | 512 (67) |
| | Mixed | 124 (16) |
| Trial start date | On or after 2015 | 210 (28) |
| | Before 2015 | 553 (72) |
| Publication link available on ClinicalTrials.gov | Yes | 148 (27) |
| | No | 403 (73) |

N: number, NA: non-available, US: United States,

[a]other includes factorial and sequential designs,

[b]other or NA: trials' primary purpose were checked to fit the previous definition of CRC management (see Methods section).

**Secondary outcome measures.** Overall, median time between primary completion date and earliest date of results availability was 32.6 months (IQ 16.1-unreached) (Fig 2). Of note, 50 interventional studies had results available before the declared primary completion date.

The cumulative percentage of interventional studies with available results (posted on the ClinicalTrials.gov registry and/or published in a full-text article) over time is shown in Fig 2 and Table 2 and was 17% at 12 months, 39% at 24 months and 55% at 36 months. Percentages according to design (randomized and non-randomized) and type of funding (industry-funded, non-industry funded and mixed funding) are shown in Fig 3A and 3B and Table 2. In univariate analysis, randomized trials and mixed-funding trials were more likely to have available results: HR = 1.3 (95% CI 1.1–1.5) and HR = 1.3 (95% CI, 1.1–1.7) respectively. In the *post-hoc* multivariate analysis, randomized trials, industry-funded trials, completed trials and trials with at least one site in the US were associated with higher results availability (Table 3).

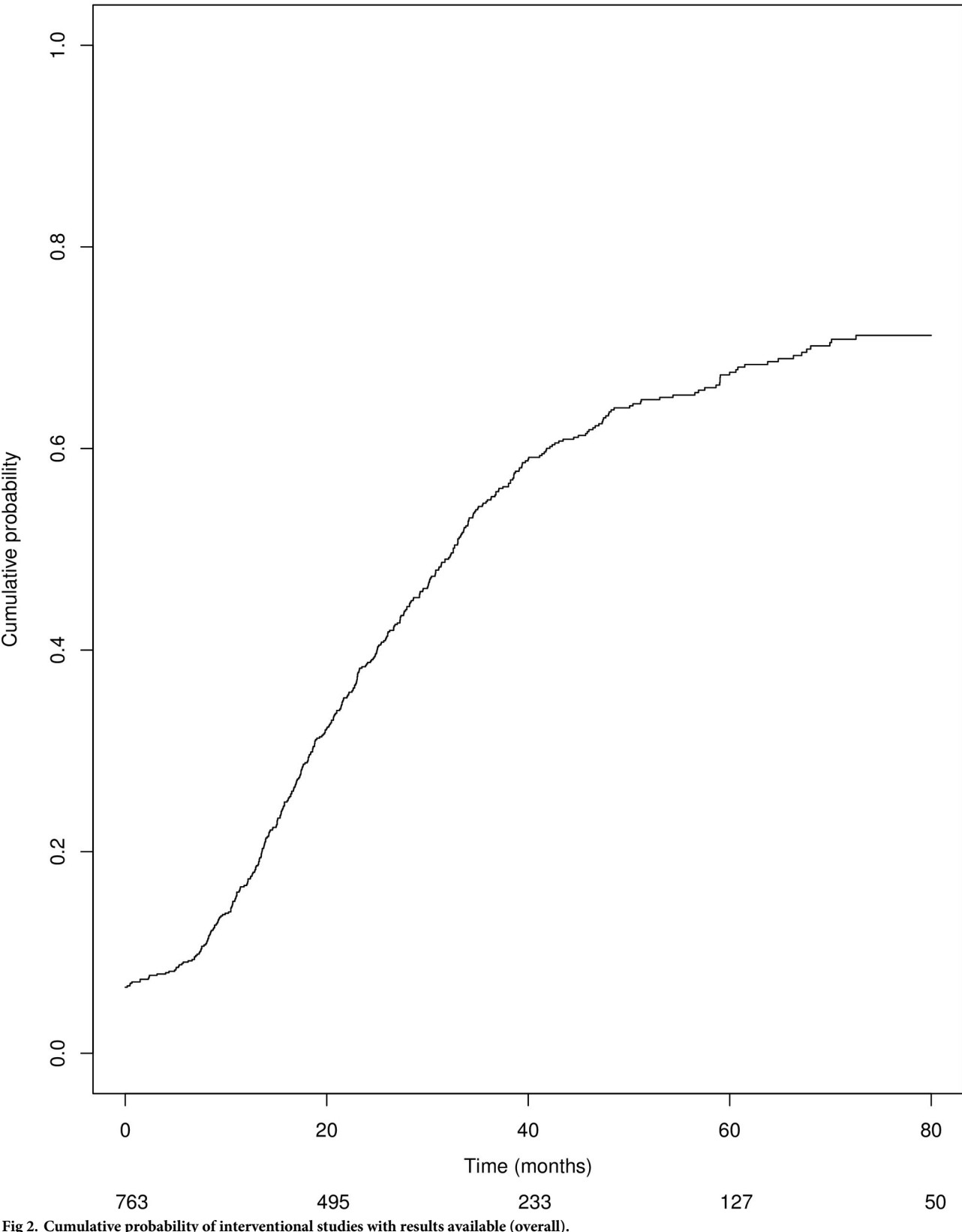

**Fig 2. Cumulative probability of interventional studies with results available (overall).**

**Table 2. Cumulative percentages of interventional studies with available or posted results over time, according to study design and type of funding.**

|  | 12 months | 24 months | 36 months |
|---|---|---|---|
| **Published and/or posted on ClinicalTrials.gov (available results)** |  |  |  |
| Overall (N = 763) | 17% | 39% | 55% |
| Design |  |  |  |
| Randomized (N = 405) | 18% | 42% | 61% |
| Non-randomized (N = 358) | 15% | 35% | 48% |
| Funding |  |  |  |
| Industry (N = 127) | 14% | 44% | 56% |
| Non-industry (N = 512) | 17% | 35% | 53% |
| Mixed (N = 124) | 20% | 46% | 60% |
| **Posted on ClinicalTrials.gov** |  |  |  |
| Overall (N = 763) | 2% | 14% | 18% |
| Design |  |  |  |
| Randomized (N = 405) | 3% | 14% | 18% |
| Non-randomized (N = 358) | 2% | 14% | 18% |
| Funding |  |  |  |
| Industry (N = 127) | 3% | 32% | 41% |
| Non-industry (N = 512) | 2% | 7% | 10% |
| Mixed (N = 124) | 3% | 22% | 26% |

N: number.

The cumulative percentage of interventional studies with posted results on the Clinical-Trials.gov registry over time is shown in Fig 4 and Table 2 and was 2% at 12 months, 14% at 24 months and 18% at 36 months. Percentages according to design (randomized and non-randomized) and type of funding (industry-funded, non-industry funded and mixed funding) are shown in Fig 5A and 5B and Table 2. In univariate analysis, both industry-funded and mixed funding trials were more likely to post their results on the registry: HR = 4.9 (95% CI 3.5–7.0) and HR = 2.9 (95% CI 2.0–4.3) respectively. Randomized trials were not more likely to post their results: HR = 1.0 (95% CI 0.8–1.4). In multivariate analysis, both industry-funded and mixed funding trials as well as trials with at least one site in the US were more likely to post their results on ClinicalTrials.gov (Table 3).

Available results were in open access in 420 (420/477 = 88%) of cases. Open access was possible through ClinicalTrials.gov and through an open access publication for 169 and 251 interventional studies respectively.

## Discussion

In our work, only 63% of interventional studies had available results over time (posted on the CinicalTrials.gov registry and/or published in a full-text article) and median time between primary completion date and date of results availability was 32.6 months (IQ 16.1-unreached). Results were more likely available for trials that were randomized, completed, had one site in the US, and industry-funded. Only 2% of CRC interventional studies posted their results on ClinicalTrials.gov within one year after the primary completion date.

Our work is the first to study results availability in a large sample of interventional studies studying CRC management. We arbitrarily looked at seven years of registered trials to get a large sample for our analysis and to allow enough time for publication.

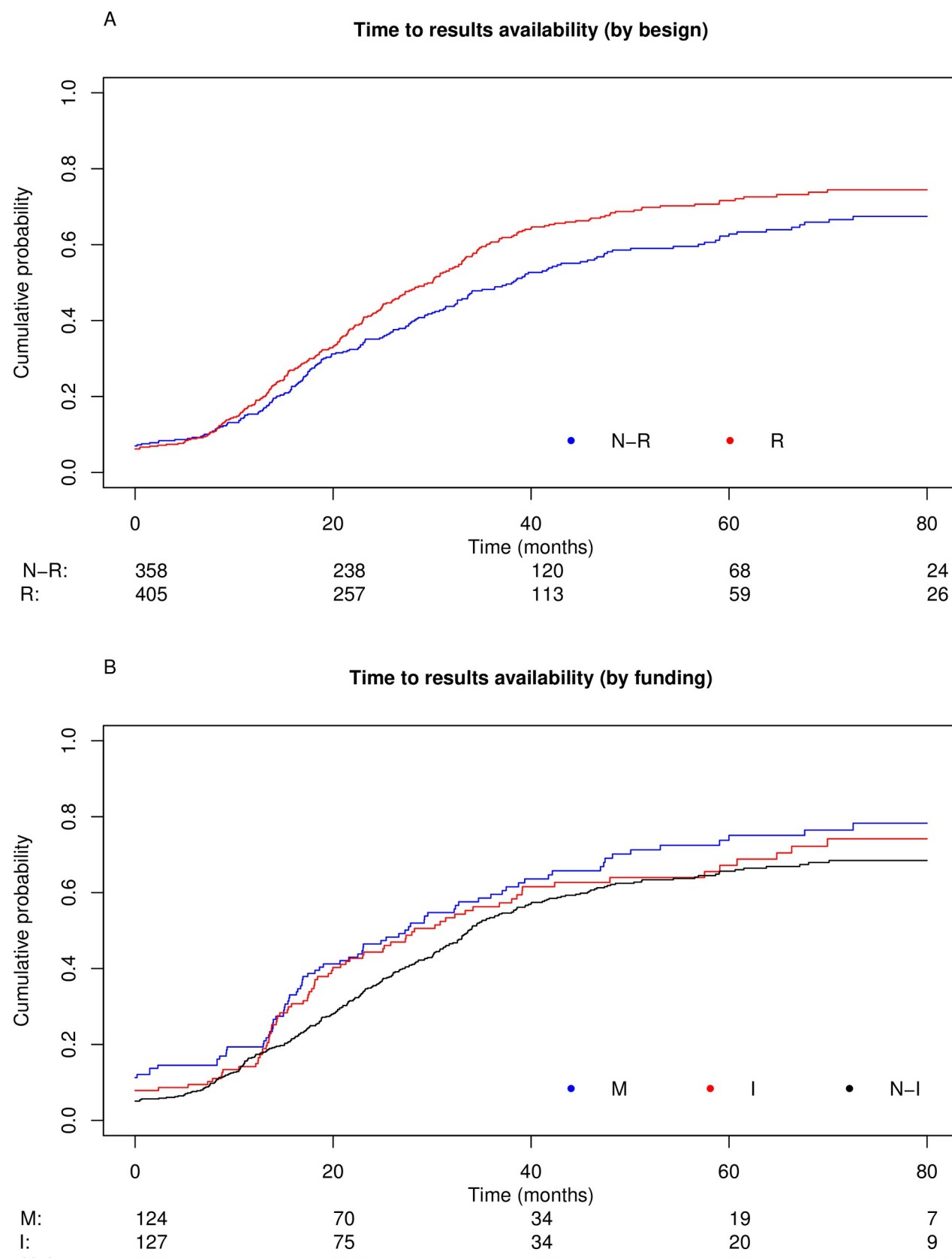

**Fig 3. Cumulative probability of interventional studies with results available, stratified by design and funding.** (A) Cumulative probability according to trial design (randomized versus non-randomized) and (B) trial funding (mixed, industry-funded, non-industry funded). N-R: non-randomized, R: randomized. M: mixed, I: industry, N-I: non-industry.

Previous works have shown that the results of many trials remain unpublished in various research fields [12–14]. Similarly, despite public registration, about 50% of randomized controlled trials (RCTs) results are never published [15]. One work published in 2013 had shown that results of nearly half of the oncology trials performed in the United States were not publicly available 3 years after completion [14]. Our results are in line with these findings, showing little improvement in the oncology field.

Availability of results is a crucial step in research. Indeed, there is a tendency to less publish trials with negative results, creating a risk of biased literature which can negatively impact future meta-analyses results and the development of therapeutic guidelines [16, 17]. Also, not publishing results raises an ethical issue for patients who agree to participate in these trials (benefit/risk balance to assess when participating in a new trial) [18]. In our work, we estimated a total of 270 845 (40%) patients involved in CRC interventional studies without available results up to eight years after the primary completion date. Our results also suggest that industry-funded trials are more likely to post or publish results, and more promptly. This is also consistent with previous findings [14, 19, 20]. Therefore, making sure that results for all trials are openly available is a priority [11, 18, 21]. Recent analysis performed on the Clinical-Trials.gov registry show an increase in the number of trials with posted results, but we cannot conclude on an improvement of trial results availability over time [22].

In our work, we focused on results published in full-text articles, not considering other supports such as abstracts. Previous evidence has suggested that the quality of oncology abstracts in oncology meetings is often suboptimal and that presented results are often primary [23, 24]. Regarding publication of results in journals, one could argue that the delay is likely worsen by journal publication processes, especially in case of multiple rejections [25]. This is why submission of results on the registry independently of journal publication is crucial to improve transparency but still underfollowed as illustrated by our findings. Indeed, some registries even require submission of results for selected trials within a defined time range after trial completion. Since 2007, the US Food and Drug Administration Amendments Act 801 (FDAAA 801) requires submission of trial results on ClinicalTrials.gov no later than one year after the

**Table 3. Trial characteristics associated with results availability and posting on ClinicalTrials.gov in multivariate analysis.**

| Trial characteristics | Availability of results | Posting on ClinicalTrials.gov |
|---|---|---|
| Design | | |
| *Randomized (vs non-randomized)* | HR = 1.4 (95% CI 1.1–1.7) | HR = 1.2 (95% CI 0.9–1.6) |
| Type of funding | | |
| *Industry (vs non-industry)* | HR = 1.4 (95% CI, 1.1–1.8) | HR = 3.2 (95% CI 2.3–4.6) |
| *Mixed (vs non-industry)* | HR = 1.1 (95% CI, 0.9–1.4) | HR = 2.5 (95% CI 1.7–3.7) |
| Trial location | | |
| *One site in the US (vs no site in the US)* | HR = 1.6 (95% CI, 1.3–1.9) | HR = 5.1 (95% CI, 3.6–7.2) |
| Trial status | | |
| *Terminated (vs completed)* | HR = 0.5 (95% CI, 0.4–0.6) | HR = 1.2 (95% CI, 0.8–1.7) |
| Study start date | | |
| *Old (vs recent)* | HR = 1.2 (95% CI, 0.9–1.5) | HR = 1.5 (95% CI, 1.0–2.3) |

CI: confidence intervals, HR: hazard ratio, US: United States, vs: versus.

**Time between primary completion and posting (overall)**

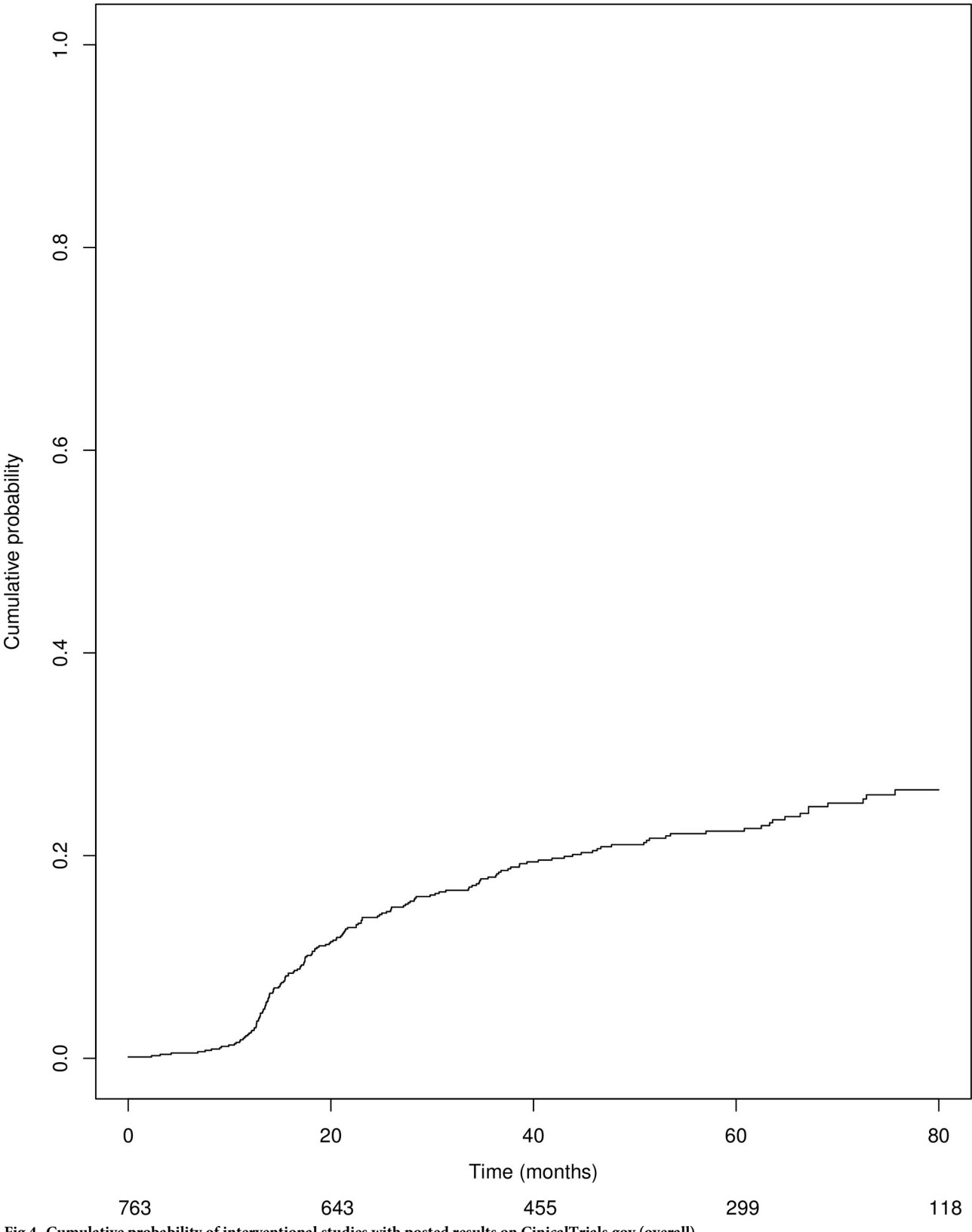

**Fig 4. Cumulative probability of interventional studies with posted results on CinicalTrials.gov (overall).**

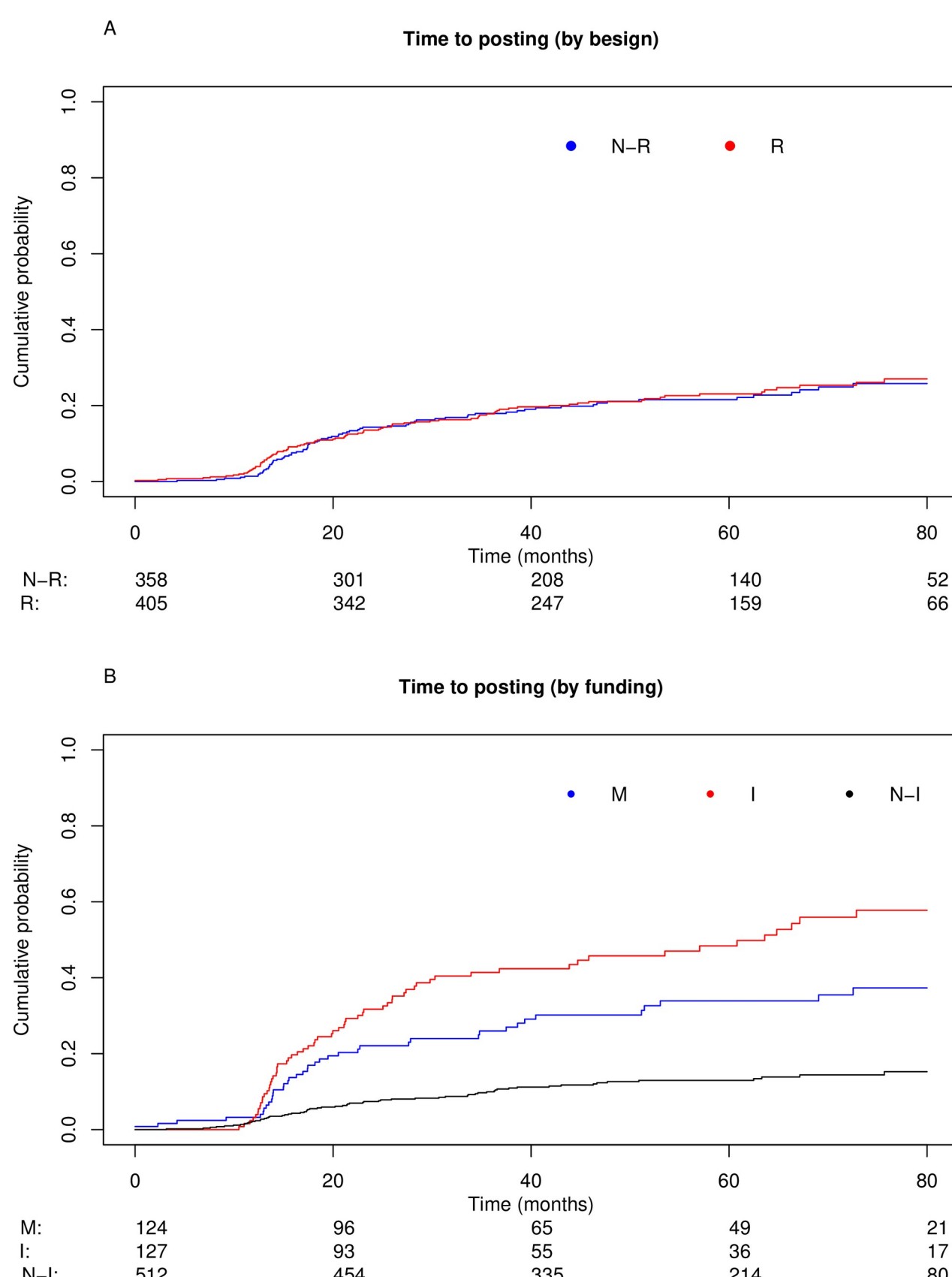

**Fig 5. Cumulative probability of interventional studies with posted results on ClinicalTrials.gov stratified by design and funding.** (A) Cumulative probability according to trial design (randomized versus non-randomized) and (B) trial funding (mixed, industry-funded, non-industry funded). N-R: non-randomized, R: randomized. I: industry, N-I: non-industry, M: mixed.

primary completion date for "applicable clinical trials" (see definition on https://www.clinicaltrials.gov/ct2/home). Similarly, for any interventional study that ended on or after 2014-07-21 and registered on the European registry (EudraCT), sponsors are requested to post trial results within 12 months following the end of the trial.

Our work has several limitations. First, we only considered interventional studies registered on ClinicalTrials.gov and cannot extrapolate to all trials. However, it is currently the largest registry and trials can be registered on more than one registry at the same time [3]. Then, the quality of information registered on the ClinicalTrials.gov registry is not always up to date and of high quality [3]. As an example, we found that 50 trials had results available before the declared primary completion date. Another explanation for this result could be the change of the definition of "primary completion date" in 2016 on ClinicalTrials.gov (it was previously unclear how to apply the definition for more than one primary outcome measure). As another example, only 53% of trials were prospectively registered in our work. Of note, US law requires trial registration within 21 days of first patient enrollment which could also partly explain this result. Also, regarding posting of results on ClinicalTrials.gov, we did not assess whether trials were considered as "applicable" according to the FDAAA 801; therefore, this result should be considered with caution as not all trials are required to post their results on the registry. Furthermore, considering the selected time range, some trials in our sample only had 14 months for assessment of results availability. Finally, some publication of results could have been missed in our work since identification has been done by one reviewer and with only one database (MEDLINE).

Many physicians from various fields, including oncologists, surgeons, pathologists, gastroenterologists, general practitioners and others, are involved in the management of CRC and should be aware of the waste of research in this field. Enhancing awareness through interventions and promotion of open research to international digestive oncology groups might help improve the transparency of future interventional studies in CRC management.

In conclusion, there is a high rate of interventional studies studying CRC management with unavailable results. There is some waste in CRC research with room for improvement.

## Supporting information

**S1 Table. Advanced search on ClinicalTrials.gov with the advanced research tool.** (DOCX)

**S2 Table. Data extraction form.** (DOCX)

## Acknowledgments

We thank Elodie Perrodeau for her help with the statistical analysis.

## Author Contributions

**Conceptualization:** Isabelle Boutron, Philippe Ravaud.

**Data curation:** Anna Pellat.

**Formal analysis:** Anna Pellat.

**Methodology:** Isabelle Boutron, Philippe Ravaud.

**Supervision:** Isabelle Boutron.

**Writing – original draft:** Anna Pellat.

**Writing – review & editing:** Anna Pellat, Isabelle Boutron, Philippe Ravaud.

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
