## [Decision Letter · Decision Letter 0]

6 Jan 2022

PONE-D-21-34103Availability of results of interventional trials assessing colorectal cancer over the past seven years.PLOS ONE

Dear Dr. Pellat,

Thank you for submitting your manuscript to PLOS ONE. After careful consideration, we feel that it has merit but does not fully meet PLOS ONE’s publication criteria as it currently stands. Therefore, we invite you to submit a revised version of the manuscript that addresses the points raised during the review process.

We look forward to receiving your revised manuscript.

Kind regards,

Tim Mathes

Academic Editor

PLOS ONE

Journal Requirements:

Reviewers' comments:

Reviewer's Responses to Questions

**Comments to the Author**

1. Is the manuscript technically sound, and do the data support the conclusions?

Reviewer #1: Partly

Reviewer #2: Partly

2. Has the statistical analysis been performed appropriately and rigorously? 

Reviewer #1: No

Reviewer #2: No

3. Have the authors made all data underlying the findings in their manuscript fully available?

Reviewer #1: Yes

Reviewer #2: Yes

4. Is the manuscript presented in an intelligible fashion and written in standard English?

Reviewer #1: No

Reviewer #2: Yes

5. Review Comments to the Author

Reviewer #1: Pellat and colleagues investigated the availability of results of interventional studies using data from the ClinicalTrials.gov registry. The authors found that the results of over a third of clinical trials are not published. The findings of the study are interesting and may have public health relevance. However, there are a number of major methodological concerns (as outlined below), which may affect the validity of the reported findings. The authors could enhance the content of the manuscript by considering the suggestions below.

Title

Trials are interventional studies. I would thus suggest that the title be changed to “interventional studies” and the term be used consistently throughout the manuscript.

Methods

• Line 80: What was the definition of CRC management for the selection of studies? Do the authors mean secondary and/or tertiary prevention? This should be clarified.

• Outcomes: Median time (rather than delay) between primary completion and earliest date of results availability. The term should be used throughout the manuscript.

• Were statistical tests one- or two-sided?

Results

• Did the primary outcome differ by geographical region, year that trials commenced, and funder type? These are important and should be evaluated in the revision.

• The 763 trials also include trials that were terminated. This is of major concern and affects the validity of the findings. Since the focus is on availability of results, trials that were terminated would certainly have no results (huge bias – like a cohort study aiming to assess incident cases but then included prevalent cases at the onset). The authors should consider this in the revision, redefine the eligibility criteria and conduct the analysis again by restricting the analyses to trials that were completed.

• Table for the Cox analysis was not shown in the manuscript and which factors were adjusted for?

Conclusions

• The first sentence in the concluding paragraph is a bit “harsh” and should be restructured.

Minor

Line 19: …most frequent cancers…

Line 35: What do the authors mean by “posted results”? Do they mean by published results?

All CI95% should be changed to 95%CI throughout the manuscript.

Line 45: …most frequent malignant neoplasms….

Lines 52-55: These sentences should be restricted for clarity.

Line 169: Continuous data….

Line 170: Binary and categorical data were….

Titles of figures 2-5 should be “Time between” and not “Delay”

Reviewer #2: The research article by Pellet A, Boutron I, and Ravaud P aims to characterize the availability of results for clinical trials of colorectal cancer management based on availability in ClinicalTrials.gov and in biomedical journals as identified through PubMed. This article reinforces information already understood in the literature about results reporting for clinical trials, but focuses on the specific domain of colorectal cancer. This research may be more appropriate for specialists in the field of colorectal cancer than a general medical audience and it is recommended that the authors clarify the need for the analysis and for whom it is intended.

General Comments

- Specify the specific years of the analysis as “past 7 years” is not a useful time reference because will change over time

- Recommend clarifying throughout ‘posted’ is ‘posted on ClinicalTrials.gov’

- Recommend phrase ‘median time’ v. ‘median delay’ as it is neutral. Delay implies something ‘wrong’ when in fact all results availability will have a delay from completion to availability

Background

- Refers to an increased number of trials conducted over time, however, this may be an overstatement as generally the information on conducted trials has increased, but it is difficult to ascertain how the number of trials themselves have increased especially to long known problems of publication bias

- Information on ‘lack of quality and standardization’ is outside the scope of this analysis and recommend authors limit background content to that which is relevant to this analysis or otherwise be clear why it is relevant background

- The description of the FDAAA 801 results requirements are oversimplified. Important qualifiers should be included: limited to Phase 2 – 4 interventional studies of FDA-regulated products. In addition, there has been shifts over time in what is expected for results reporting of unapproved products (e.g., trials of unapproved products are expected no later than 3 years after PCD if unapproved or within 30 days of product approval)

Methods

- The definition of “CRC management” and how this is applied is unclear. Recommend the authors include more details in the supplemental appendix that would allow for replication of search strategy and application of exclusionary criteria (“e.g.,” on p.5 line 107 does not seem sufficient). I also question why the Primary Purpose data element was not used to exclude ‘screening, basic science, device feasibility, etc.’ type studies that might not be considered management.

- The definition of ‘primary completion date’ was clarified in 2016 and the clarified definition implemented in 2017 (it was previous unclear how to apply if more than one primary outcome measure). This sample includes trials that cross this definitional time point and this may be helpful to note when providing the definition as may also partially explain why some results available before primary completion date.

- Eligibility indicates inclusion was 16 years old and over, but search methods only included 18 years old and over per Table S1 (also confusing why exclude “performed in children” based on the search if the trials still included adults

- Primary completion date max of 01/01/2020 means only 14 months since that date for some studies; recommend reporting in results what the median/range of time from primary completion at time of sample (impacts results available)

- Define ‘results posted’ (excludes ‘submitted results with QC comments available’)

- All trials without available results censored on April 15th, 2021 – mentioned in relation to publication but not posted, does it apply to that category as well?

- Define ‘main results’ – are these results of the trial’s primary outcome measure?

- Explain rationale for stratification choices

- Define ‘open access’ – results in ClinicalTrials.gov OR open access publication?

- The statistical methods that apply confidence intervals do not appear to be the best choice for this analysis as the data are descriptive of the results available for trials registered in ClinicalTrials.gov and are not an inference to a larger population. The sample is the population. For additional information on this topic, see “Use of appropriate statistical inference” and “How can I objectively identify important differences?” described here: https://aact.ctti-clinicaltrials.org/points_to_consider

Results - Table 1

- Consider adding some key facts mentioned in ‘general characteristics of trials’ included ‘open-labeled’ and related categories

- As noted in methods, unclear if/how ‘screening, other, NA’ qualifies as ‘CRC management’

- Need to define ‘type of funding’ and publication link posted in registry’ with Table or in Methods or supplemental

- Suggest including extracted information related to location to characterize the study sample based on location

Results

- ‘prospectively registered’ – note U.S. law requires registration within 21 days of first patient enrollment, the definition used is acceptable just can’t infer anything per U.S. law

- Page 11 (line 212) – results re: ‘another support’ appear to be out of scope from this analysis as the Methods only described full-text articles

- As noted in methods, the use of confidence intervals does not seem to be appropriate for this analysis; please re-evaluate throughout

- Page 11 (line 232) – unclear to what industry-funded trials is being compared

- Table 2 would benefit from including a line for ‘all published or posted’ and ‘all posted’ to complement the information

- Consider optimal use of Figures and Tables as they currently appear to included redundant information

- Methods/Results – clarify what proportion of results were published as open access

- Figures need additional labels for the numbers (of trials) provided with KM graph. Also consider if y-axis should include full scale of probability 0.0 – 1.0 for accurate context.

Discussion - the discussion needs significant attention to better focus on and reflect the results from this analysis and their relevance to the colorectal cancer community

- The discussion (page 13, line 268) discusses ‘compliance’ however the objectives and methods of this paper do not support any inference about compliance and this should be deleted (i.e., analysis was not limited to phase 2-4 trials of FDA-regulated drug products and did not assess other appropriate reasons for results being available later than 12 months after completion)

- Page 14, line 274 specifies ‘enough time for publication’ however as noted earlier some studies had only 14 months; consider specifying as limitation

- Page 14, line 275 mentions ‘the announcement’ – should specify who made this announcement; this discussion point is disjointed from the background that did not mention the ICMJE policy at all; recommend bringing the background and discussion into alignment

- I expect there are additional references beyond reference 13 that could be sued to support the sentence ending on line 281 (page 14)

- Page 14, line 286 – specify what the ethical issue is

- Page 14, lines 291 -3. Unclear why IPD and ‘study documentations’ are being discussed as this was not a focus of this paper.

- Reference 21 is not appropriate for supporting the statement on page 14, lines 293-295 as it shows more results in ClinicalTrials.gov but can’t characterize as an ‘improvement’

Minor editorial

- Section headers of ‘search strategy for posted results’ and ‘search strategy for publication of results’ seem unnecessary, could be combined

- ‘overtime’ should be ‘over time’

6. PLOS authors have the option to publish the peer review history of their article (what does this mean?). If published, this will include your full peer review and any attached files.

Reviewer #1: No

Reviewer #2: No

---

## [Author Response · Author response to Decision Letter 0]

20 Jan 2022

Dear Editors, 

We thank you and the two anonymous referees for the detailed and constructive comments on our manuscript entitled “Availability of results of interventional studies assessing colorectal cancer over the past seven years” by Pellat et al. Please find enclosed a revised version that we submit for publication that takes these new comments into account. You will find below point-by-point responses to the comments and description of the changes made in the manuscript.

Thank you for your attention on our revised manuscript.

Sincerely yours,

Ms. Anna Pellat

 

Reviewer comments: 

Reviewer #1 

Pellat and colleagues investigated the availability of results of interventional studies using data from the ClinicalTrials.gov registry. The authors found that the results of over a third of clinical trials are not published. The findings of the study are interesting and may have public health relevance. However, there are a number of major methodological concerns (as outlined below), which may affect the validity of the reported findings. The authors could enhance the content of the manuscript by considering the suggestions below.

*Comment 1: 

Title: Trials are interventional studies. I would thus suggest that the title be changed to “interventional studies” and the term be used consistently throughout the manuscript.

Response 1: 

Thank you for this suggestion. We have made the changes as suggested in the revised version of the manuscript by keeping both terms “interventional studies” and “trials”.

*Comment 2:

Methods:

• Line 80: What was the definition of CRC management for the selection of studies? Do the authors mean secondary and/or tertiary prevention? This should be clarified.

Response 2:

This is an important comment. We defined as management all trials focusing on screening, diagnosis, prevention (secondary and tertiary), treatment (including drugs, dietary supplement, behavioral intervention, device and surgery), and supportive care. When the primary purpose of the study was not specified (“other” or not available) we checked that each trial’s primary purpose fitted our previous definition. For more clarity, we have made the following changes in the revised version of the manuscript:

Methods section, eligibility criteria, page 5:

We defined as CRC management all trials focusing on screening, diagnosis, prevention (secondary and tertiary), treatment (including drugs, dietary supplement, behavioral intervention, device and surgery), and supportive care. When the primary purpose of the study was not specified (“other” or not available) we checked that each trial’s primary purpose fitted our previous definition.

*Comment 3:

• Outcomes: Median time (rather than delay) between primary completion and earliest date of results availability. The term should be used throughout the manuscript.

Response 3:

Thank you for this suggestion. We have made the change in the revised version of the manuscript and in the four figures.

*Comment 4:

• Were statistical tests one- or two-sided?

Response 4:

Thank you for this comment. Following the other reviewer’s comments on statistical analysis, we have modified the statistical analysis section of the revised manuscript as follows:

Statistical analysis section, page 9:

“We used the R software (R studio Version 1.2.5033) for all statistical analysis. Continuous data were given in mean with standard deviation (SD), or median with interquartile (IQ) range. Binary and categorical data were given in percentages. 

For the assessment of cumulative percentages and time to event analysis we used the Kaplan Meier method. If the date of publication of results was anterior to the primary completion date (negative time for survival), we considered that results were published on the date of trial completion. 

We estimated the hazard ratios (HR) by using the Cox Proportional-Hazards model to evaluate the effect of two pre-specified trial characteristics on time to event (type of funding and trial design). We also performed a post-hoc multivariate analysis to evaluate three additional trial characteristics: trial location, trial status and trial start date. We tested the proportional hazard assumption for each variable in our model by using the cox.zph function on R and plotting the scaled Schoenfeld residuals against time. 

Because the data set included all studies in the population of interest, analytical methods were descriptive. In accordance with guidance provided by CTTI, statistical inference was not performed. Therefore, 95% confidence intervals (CI) contained in this paper represent population parameters rather than sample statistics.”

*Comment 5:

Results:

• Did the primary outcome differ by geographical region, year that trials commenced, and funder type? These are important and should be evaluated in the revision.

Response 5: 

Thank you for this question. In our initial version of the manuscript we had only evaluated in univariate analysis if the outcome differed by two pre-specified trial characteristics: the type of funding and the trial design (randomized or non-randomized). As requested, we will perform a post-hoc multivariate analysis with new trial characteristics: trial location (one site in the US versus other), trial start date and trial status (terminated or completed).

We have added a paragraph in the methods section and made the following changes in the revised version of the manuscript:

Methods section, Trial characteristics evaluated for association with results availability, page 7:

“To examine the association of trial characteristics with availability of results and posting of results on ClinicalTrials.gov we a priori selected two explanatory variables based on their interest following previous findings in the literature: type of funding and trial design. Type of funding was divided in three categories: industry-funded, non-industry funded (e.g. academic funding) and mixed funding (both industry and non-industry funding). Trial design was divided in two categories: randomized versus non-randomized. We also performed a post-hoc analysis to evaluate the effect of three additional trial characteristics: trial location (at least one trial site in the US versus no trial site in the US), trial status and trial start date. For evaluation of the trial start date, we compared recent trials with a start date on or after 2015-01-01 with older trials (before 2015).”

Statistical analysis section, page 9:

“We also performed a post-hoc multivariate analysis to evaluate three additional trial characteristics: trial location, trial status and trial start date.”

Results section, New table 3, page 15:

Table 3. Trial characteristics associated with results availability and posting on ClinicalTrials.gov in multivariate analysis.

Trial characteristics Availability of results Posting on ClinicalTrials.gov

Design

Randomized (vs non-randomized) 

HR= 1.4 (95% CI 1.1-1.7) 

HR=1.2 (95% CI 0.9-1.6)

Type of funding

Industry (vs non-industry)

Mixed (vs non-industry) 

HR=1.4 (95% CI, 1.1-1.8) 

HR = 1.1 (95% CI, 0.9-1.4) 

HR= 3.2 (95% CI 2.3-4.6) 

HR=2.5 (95% CI 1.7-3.7) 

Trial location

One site in the US (vs no site in the US) 

HR= 1.6 (95% CI, 1.3-1.9) 

HR=5.1 (95% CI, 3.6-7.2) 

Trial status

Terminated (vs completed) 

HR=0.5 (95% CI, 0.4-0.6) 

HR=1.2 (95% CI, 0.8-1.7)

Study start date

Old (vs recent) 

HR=1.2 (95% CI, 0.9-1.5) 

HR=1.5 (95% CI, 1.0-2.3)

CI: confidence intervals, HR: hazard ratio, US: United States, vs: versus

*Comment 6:

Results 

• The 763 trials also include trials that were terminated. This is of major concern and affects the validity of the findings. Since the focus is on availability of results, trials that were terminated would certainly have no results (huge bias – like a cohort study aiming to assess incident cases but then included prevalent cases at the onset). The authors should consider this in the revision, redefine the eligibility criteria and conduct the analysis again by restricting the analyses to trials that were completed.

Response 6:

Thank you for this comment. We respectfully disagree that we should expect no results from terminated trials. Indeed, in our sample, 60 out of 142 terminated trials had available results (posted on ClinicalTrials.gov and/or published in a full-text article). Therefore, we considered that excluding them would also create a bias. We do agree that is more likely for completed trials to have more results available than terminated trials. To evaluate the influence of that characteristic on results availability in our sample, we performed a post-hoc analysis by adding trial status (see previous comment/response 5). 

*Comment 7:

• Table for the Cox analysis was not shown in the manuscript and which factors were adjusted for?

Response 7:

Thank you for this comment. In our initial version of the manuscript we only evaluated two pre-specified trial characteristics in univariate analysis, so without adjustment.

Following the reviewer’s request, we have added a post-hoc multivariate analysis with three additional trial characteristics (study location, trial status and study start date), and results were presented in the new Table 3 (see previous response 5).

*Comment 8:

Conclusions:

• The first sentence in the concluding paragraph is a bit “harsh” and should be restructured.

Response 8:

Thank you for this suggestion. We have changed the conclusion in the revised version of the manuscript as follows:

Conclusion section, page 17:

“In conclusion, there is a high rate of interventional studies studying CRC management with unavailable results. There is some waste in CRC research with room for improvement.”

*Comment 9:

Minor:

Line 19: …most frequent cancers…

Line 35: What do the authors mean by “posted results”? Do they mean by published results?

All CI95% should be changed to 95%CI throughout the manuscript.

Line 45: …most frequent malignant neoplasms….

Lines 52-55: These sentences should be restricted for clarity.

Line 169: Continuous data….

Line 170: Binary and categorical data were….

Titles of figures 2-5 should be “Time between” and not “Delay”

Response 9:

Thank you for these rectifications. We have made all the changes in the revised version of the manuscript as well as in the figures.

Of note, “Posted results” means “posted on the ClinicalTrials.gov registry”, irrespective of publication in a full-text article.

Reviewer #2

The research article by Pellat A, Boutron I, and Ravaud P aims to characterize the availability of results for clinical trials of colorectal cancer management based on availability in ClinicalTrials.gov and in biomedical journals as identified through PubMed. This article reinforces information already understood in the literature about results reporting for clinical trials, but focuses on the specific domain of colorectal cancer. 

*Comment 1: 

This research may be more appropriate for specialists in the field of colorectal cancer than a general medical audience and it is recommended that the authors clarify the need for the analysis and for whom it is intended.

Response 1:

Thank you for this important comment. Colorectal cancer being an important and frequent cancer, and one of the few cancers for which screening is available, many actors are involved in its management (digestive oncologists, oncologists, surgeons, pathologists, gastroenterologists, general practitioners, epidemiologists…). Therefore, it is important for all of these actors to be aware of the waste in research in this field. In this sense, we think that is work is suited for a general medical audience. We have added a sentence in the discussion section of the revised manuscript as follows:

Discussion section, page 17:

“Many physicians from various fields, including oncologists, surgeons, pathologists, gastroenterologists, general practitioners and others, are involved in the management of CRC and should be aware of the waste of research in this field.”

*Comment 2: 

General Comments

- Specify the specific years of the analysis as “past 7 years” is not a useful time reference because will change over time

- Recommend clarifying throughout ‘posted’ is ‘posted on ClinicalTrials.gov’

- Recommend phrase ‘median time’ v. ‘median delay’ as it is neutral. Delay implies something ‘wrong’ when in fact all results availability will have a delay from completion to availability

Response 2:

Thank you for these suggestions. We have made all these changes in the revised version of the manuscript.

*Comment 3:

Background

- Refers to an increased number of trials conducted over time, however, this may be an overstatement as generally the information on conducted trials has increased, but it is difficult to ascertain how the number of trials themselves have increased especially to long known problems of publication bias

- Information on ‘lack of quality and standardization’ is outside the scope of this analysis and recommend authors limit background content to that which is relevant to this analysis or otherwise be clear why it is relevant background

Response 3:

The reviewer is correct. We have made the following changes in the background section of the revised manuscript:

Background section, page 3:

“Since the 2004 announcement by the International Committee of Medical Journal Editors of required registration for each new clinical trial before participant enrollment and as a condition for publication, there has been a strong increase of new trial registrations worldwide with a better identification of the ongoing research (1–3). The ClinicalTrials.gov registry, developed by the United states (US) National Institutes of Health (NIH), is currently the largest used registry (1,3). In the oncology field, there has been an increased number of conducted interventional studies over time, including a growing number of pilot trials and industry-sponsored trials (4–6). As a consequence, cancer drugs have comprised the single largest category of new drug approvals in Europe in the recent years (6).

However, previous works have highlighted an important waste in the production and reporting of research in various fields because of lack of quality and standardization in the different research steps (7–9). Among others, this waste could be related to inaccessible research results with a risk of biased evidence and literature (7,9–11). Trial results can be accessible through registries and/or published in various supports (abstracts, full-text articles, summary reports…). The aim of our work was to assess the availability of results for interventional CRC studies registered on the ClinicalTrials.gov registry and completed or terminated between 01/01/2013 and 01/01/2020.” 

*Comment 4:

- The description of the FDAAA 801 results requirements are oversimplified. Important qualifiers should be included: limited to Phase 2 – 4 interventional studies of FDA-regulated products. In addition, there has been shifts over time in what is expected for results reporting of unapproved products (e.g., trials of unapproved products are expected no later than 3 years after PCD if unapproved or within 30 days of product approval)

Response 4:

We agree with the reviewer. After revision, we think that mentioning the FDAAA801 rule of posting results for applicable trials in the background is irrelevant as it is not the main objective of our work. We have deleted the paragraph in the introduction and only mentioned the FDAAA 801 in the discussion section with more precision as requested.

Discussion section, page 18:

“This is why submission of results on the registry independently of journal publication is crucial to improve transparency but still underfollowed as illustrated by our findings. Indeed, some registries require submission of results for selected trials within a defined time range after trial completion. Since 2007, the US Food and Drug Administration Amendments Act 801 (FDAAA 801) requires submission of trial results on ClinicalTrials.gov no later than one year after the primary completion date for applicable trials (see definition on https://www.clinicaltrials.gov/ct2/home). Similarly, for any interventional study that ended on or after 2014-07-21 and registered on the European registry (EudraCT), sponsors are requested to post trial results within 12 months following the end of the trial.”

*Comment 5:

Methods

- The definition of “CRC management” and how this is applied is unclear. Recommend the authors include more details in the supplemental appendix that would allow for replication of search strategy and application of exclusionary criteria (“e.g.,” on p.5 line 107 does not seem sufficient). I also question why the Primary Purpose data element was not used to exclude ‘screening, basic science, device feasibility, etc.’ type studies that might not be considered management.

Response 5:

This is an important comment which was also made by the other reviewer. We defined as management all trials focusing on screening, diagnosis, prevention (secondary and tertiary), treatment (including drugs, dietary supplement, behavioral intervention, device and surgery), and supportive care. In our sense, screening is an important part of CRC management. 

When the primary purpose of the study was not specified (“other” or not available) we checked that each trial’s primary purpose fitted our previous definition. One trial mentioned a “device feasibility” purpose on Cinicaltrials.gov, which was actually a diagnostic procedure in our view. Similarly, trials with a “basic science” primary purpose actually had treatment, diagnostic or prevention purposes. 

For more clarity, we have made the following changes in the revised version of the manuscript and in Table 1:

Methods section, eligibility criteria, page 5:

“We defined as CRC management all trials focusing on screening, diagnosis, prevention (secondary and tertiary), treatment (including drugs, dietary supplement, behavioral intervention, device and surgery), and supportive care. When the primary purpose of the study was not specified (“other” or not available) we checked that each trial’s primary purpose fitted our previous definition.”

Results section, Table 1, page 10:

Table 1. General characteristics of the eligible interventional studies (N=763).

Characteristics of interventional studies N, (%)

Status 

 Terminated

Completed 142 (19)

621 (81)

Study allocation

 Randomized

Non-randomized 405 (53)

358 (47)

Study design

 Single group

Parallel

Cross over

Othera or NA 321 (42)

403 (53)

19 (2)

20 (3)

Blinding Yes (any type)

None (open label)

NA 173 (22)

585 (77)

5 (1)

Primary purpose

 Screening

Prevention

Diagnostic

Treatment

Supportive care

Other or NAb 69 (9)

56 (7)

53 (7)

505 (66)

47 (6)

33 (5)

Enrollment type

 Actual

Estimated or NA 734 (96)

28 (4)

Trial location At least one site in the US

No site in the US 284 (37)

479 (63)

Type of funding 

 Industry

Non-industry

Mixed 127 (17)

512 (67)

124 (16)

Trial start date On or after 2015

Before 2015 210 (28)

553 (72)

Publication link available on ClinicalTrials.gov

 Yes

No 148 (27)

403 (73)

N: number, NA: non-available, US: United States, aother includes factorial and sequential designs, bother or NA: trials’ primary purpose were checked to fit the previous definition of CRC management (see Methods section).

*Comment 6:

- The definition of ‘primary completion date’ was clarified in 2016 and the clarified definition implemented in 2017 (it was previous unclear how to apply if more than one primary outcome measure). This sample includes trials that cross this definitional time point and this may be helpful to note when providing the definition as may also partially explain why some results available before primary completion date.

Response 6:

Thank you for this important remark. We have used the definition as posted on the registry on the 11th of March 2021. We have added this information in the Methods section of the revised manuscript and made the following changes in the discussion section:

Methods section, eligibility criteria, page 5:

“As stated on ClinicalTrials.gov on the 11th of March 2021, primary completion date was defined as the date on which the last participant in the study was examined or received an intervention to collect final data for the primary outcome measure…”

Discussion section, page 17:

“Also, the quality of information registered on the ClinicalTrials.gov registry is not always up to date and of high quality (3). As an example, we found that 50 trials had results available before the declared primary completion date. Another explanation for this result could be the change of the definition of “primary completion date” in 2016 on ClinicalTrials.gov (it was previously unclear how to apply the definition for more than one primary outcome measure).”

*Comment 7:

- Eligibility indicates inclusion was 16 years old and over, but search methods only included 18 years old and over per Table S1 (also confusing why exclude “performed in children” based on the search if the trials still included adults

Response 7:

The reviewer is correct and we have corrected our mistake. Inclusion was 16 years old and over. We made the changes in Table S1 and deleted “performed in children” in the revised version of the manuscript in the Methods section.

*Comment 8:

- Primary completion date max of 01/01/2020 means only 14 months since that date for some studies; recommend reporting in results what the median/range of time from primary completion at time of sample (impacts results available)

Response 8:

Thank you for this comment. In the initial version of the manuscript we had indeed reported the median time between primary completion date and earliest date of results availability for our whole sample (secondary outcome). We have also kept this result in the revised version of the manuscript : “Overall, median time between primary completion date and earliest date of results availability was 32.6 months (IQ 16.1-unreached).”

*Comment 9:

- Define ‘results posted’ (excludes ‘submitted results with QC comments available’)

Response 9:

Thank you for this important suggestion. We have updated the definition of “results posted” in the revised version of the manuscript. We have made the changes as follows:

Methods section, eligibility criteria, page 7:

“Regarding results posted on the ClinicalTrials.gov registry, we only considered fully accessible final results (any type of results). We did not include submitted results undergoing quality control check by ClinicalTials.gov.”

*Comment 10:

- All trials without available results censored on April 15th, 2021 – mentioned in relation to publication but not posted, does it apply to that category as well?

Response 10:

Indeed, this sentence only applies to publication of the results in a full-text article but not to posted results. We have made the following changes in the revised version of the manuscript:

Methods section, identification of results availability, page 7:

“All interventional studies without results in a full-text article were censored on April 15th, 2021.”

*Comment 11:

- Define ‘main results’ – are these results of the trial’s primary outcome measure?

Response 11:

Thank you for this important comment. In our work we considered any type of results and not only primary outcome results. For a better understanding, we have made the following changes in the revised manuscript.

Methods section, eligibility criteria, page 7:

“Regarding results posted on the ClinicalTrials.gov registry, we only considered fully accessible final results (any type of results). We did not include submitted results undergoing quality control check by ClinicalTials.gov.

Regarding online publication of results outside of the registry, we considered publications reporting any type of results for the interventional study.”

*Comment 12:

- Explain rationale for stratification choices

Response 12:

Thank you for this question. We had a priori selected two pre-specified trial characteristics that we considered important following previous results in the literature. Indeed, as described in our discussion section, industry-funded results were previously found more likely to report trial results. Similarly, we considered that randomized controlled trials being the gold standard of interventional studies, it could impact availability of results. For a better understanding, we have added a new paragraph and made the following changes in the revised manuscript:

Methods section, Trial characteristics evaluated for association with results availability, page 7:

“To examine the association of trial characteristics with availability of results and posting of results on ClinicalTrials.gov we a priori selected two explanatory variables based on their interest following previous findings in the literature: type of funding and trial design. Type of funding was divided in three categories: industry-funded, non-industry funded (e.g. academic funding) and mixed funding (both industry and non-industry funding). Trial design was divided in two categories: randomized versus non-randomized. We also performed a post-hoc analysis to evaluate the effect of three additional trial characteristics: trial location (at least one trial site in the US versus no trial site in the US), trial status and trial start date. For evaluation of the trial start date, we compared recent trials with a start date on or after 2015-01-01 with older trials (before 2015).”

*Comment 13:

- Define ‘open access’ – results in ClinicalTrials.gov OR open access publication?

Response 13:

Thank you for this comment. Open access was defined as accessible on ClinicalTrials.gov and/or in an open access publication downloaded from our institution. We have updated the definition in the revised version of the manuscript:

Methods section, secondary outcome measures, page 9:

“The percentage of results available in open access (on ClinicalTrials.gov and/or in an open access publication downloaded from our institution).”

*Comment 14:

- The statistical methods that apply confidence intervals do not appear to be the best choice for this analysis as the data are descriptive of the results available for trials registered in ClinicalTrials.gov and are not an inference to a larger population. The sample is the population. For additional information on this topic, see “Use of appropriate statistical inference” and “How can I objectively identify important differences?” described here: https://aact.ctti-clinicaltrials.org/points_to_consider

Response 14:

Thank you for this very important comment. We have looked into the documentation on the https://aact.ctti-clinicaltrials.org/points_to_consider. After revision with our statisticians, and following the reviewer’s comment, we have changed our statistical analysis as follows. Cumulative probabilities were presented without CI and studied differences with the Cox Proportional-Hazards model.

Statistical analysis section, page 9:

“We used the R software (R studio Version 1.2.5033) for all statistical analysis. Continuous data were given in mean with standard deviation (SD), or median with interquartile (IQ) range. Binary and categorical data were given in percentages. 

For the assessment of cumulative percentages and time to event analysis we used the Kaplan Meier method. If the date of publication of results was anterior to the primary completion date (negative time for survival), we considered that results were published on the date of trial completion. 

We estimated the hazard ratios (HR) by using the Cox Proportional-Hazards model to evaluate the effect of two pre-specified trial characteristics on time to event (type of funding and trial design). We also performed a post-hoc multivariate analysis to evaluate three additional trial characteristics: trial location, trial status and trial start date. We tested the proportional hazard assumption for each variable in our model by using the cox.zph function on R and plotting the scaled Schoenfeld residuals against time. 

Because the data set included all studies in the population of interest, analytical methods were descriptive. In accordance with guidance provided by CTTI, statistical inference was not performed. Therefore, 95% confidence intervals (CI) contained in this paper represent population parameters rather than sample statistics.”

Results section, New table 3, page 15:

Table 3. Trial characteristics associated with results availability and posting on ClinicalTrials.gov in multivariate analysis.

Trial characteristics Availability of results Posting on ClinicalTrials.gov

Design

Randomized (vs non-randomized) 

HR= 1.4 (95% CI 1.1-1.7) 

HR=1.2 (95% CI 0.9-1.6)

Type of funding

Industry (vs non-industry)

Mixed (vs non-industry) 

HR=1.4 (95% CI, 1.1-1.8) 

HR = 1.1 (95% CI, 0.9-1.4) 

HR= 3.2 (95% CI 2.3-4.6) 

HR=2.5 (95% CI 1.7-3.7) 

Trial location

One site in the US (vs no site in the US) 

HR= 1.6 (95% CI, 1.3-1.9) 

HR=5.1 (95% CI, 3.6-7.2) 

Trial status

Terminated (vs completed) 

HR=0.5 (95% CI, 0.4-0.6) 

HR=1.2 (95% CI, 0.8-1.7)

Study start date

Old (vs recent) 

HR=1.2 (95% CI, 0.9-1.5) 

HR=1.5 (95% CI, 1.0-2.3)

CI: confidence intervals, HR: hazard ratio, US: United States, vs: versus

*Comment 15:

Results - Table 1

- Consider adding some key facts mentioned in ‘general characteristics of trials’ included ‘open-labeled’ and related categories

Response 15:

Thank you for this suggestion. As suggested, we have added the “blinding” category 

as well as the “trial location” category and the “trial start date” category in Table 1 of the manuscript as follows (see response 5)

*Comment 16:

- As noted in methods, unclear if/how ‘screening, other, NA’ qualifies as ‘CRC management’

Response 16:

Thank you for this comment. We have previously answered it in comment/response 5.

*Comment 17:

- Need to define ‘type of funding’ and publication link posted in registry’ with Table or in Methods or supplemental

Response 17:

Thank you for this important remark. Definition of “publication link posted in registry” was updated in the Methods section and definition of “type of funding” in the new paragraph “trial characteristics evaluated for association with results availability” of the revised manuscript.

Methods section, identification of results availability, page 6:

“To identify publication of results, we used the publication link in ClinicalTrials.gov when available (direct access to PubMed publication).”

Methods section, Trial characteristics evaluated for association with results availability, page 7:

“To examine the association of trial characteristics with availability of results and posting of results on ClinicalTrials.gov we a priori selected two explanatory variables based on their interest following previous findings in the literature: type of funding and trial design. Type of funding was divided in three categories: industry-funded, non-industry funded (e.g. academic funding) and mixed funding (both industry and non-industry funding). Trial design was divided in two categories: randomized versus non-randomized. We also performed a post-hoc analysis to evaluate the effect of three additional trial characteristics: trial location (at least one trial site in the US versus no trial site in the US), trial status and trial start date. For evaluation of the trial start date, we compared recent trials with a start date on or after 2015-01-01 with older trials (before 2015).”

*Comment 18:

- Suggest including extracted information related to location to characterize the study sample based on location

Response 18:

Thank you for this comment. As mentioned in responses 5 and 14 we have added the « trial location » category in Table 1: trials with at least one site in the US versus no site in the US. We have also studied that characteristic in the post-hoc multivariate analysis.

See Modification in Table 1 and previous comments on statistical analysis.

*Comment 19:

Results

- ‘prospectively registered’ – note U.S. law requires registration within 21 days of first patient enrollment, the definition used is acceptable just can’t infer anything per U.S. law

Response 19:

The reviewer is correct and we thank him for this precision. In our work we have defined “prospectively registered” as “registered before the study start date” with no reference to US law. We have updated the discussion in the revised version of the manuscript:

Discussion section, page 18:

“As another example, only 53% of trials were prospectively registered in our work.

Of note, US law requires trial registration within 21 days of first patient enrollment which could also partly explain this result.”

*Comment 20:

- Page 11 (line 212) – results re: ‘another support’ appear to be out of scope from this analysis as the Methods only described full-text articles

Response 20:

The reviewer is correct. We have deleted any reference to “other support” in the methods section and results section of the revised manuscript.

*Comment 21:

- As noted in methods, the use of confidence intervals does not seem to be appropriate for this analysis; please re-evaluate throughout

Response 21:

Thank you for this comment. Indeed, we have previously answered it in comment 14/response 14. 

*Comment 22:

- Page 11 (line 232) – unclear to what industry-funded trials is being compared

Response 22:

Thank you for this comment. Industry-funded trials and mixed-funded trials were each separately compared to non-industry funded trials. 

*Comment 23:

- Table 2 would benefit from including a line for ‘all published or posted’ and ‘all posted’ to complement the information

Response 23:

Thank you for this suggestion. We have added the information in the new version of Table 2.

Results section, Table 2, page 13:

Table 2. Cumulative percentages of interventional studies with available or posted results over time, according to study design and type of funding.

 12 months 24 months 36 months

Published and/or posted on ClinicalTrials.gov (available results)

Overall 17% 39% 55% 

Design

Randomized

Non-randomized 18% 

15% 42% 

35% 61% 

48% 

Funding

Industry

Non-industry

Mixed 14% 

17% 

20% 44% 

35% 

46% 56% 

53% 

60% 

Posted on ClinicalTrials.gov

Overall 2% 14% 18% 

Design

Randomized

Non-randomized 3% 

2% 14% 

14% 18% 

18% 

Funding

Industry

Non-industry 

Mixed 3% 

2% 

3% 32% 

7% 

22% 41% 

10% 

26% 

*Comment 24:

- Consider optimal use of Figures and Tables as they currently appear to include redundant information

Response:

Thank you for this comment. We think that keeping all figures and the new Table 2 is important for more clarity and to facilitate reading. Survival curves give a better idea of what happens after 3 years since primary completion date.

*Comment 25:

- Methods/Results – clarify what proportion of results were published as open access

Response 25:

Thank you for this comment. First, as previously requested (see comment/response 13) we have updated the definition of open access. 

Result for open access was already mentioned in the initial version of the manuscript and was updated in the revised version as follows: 

Results section, page 16:

“Available results were in open access in 420 (420/477=88%) of cases. Open access was possible through ClinicalTrials.gov and through an open access publication for 169 and 251 interventional studies respectively. “

*Comment 26:

- Figures need additional labels for the numbers (of trials) provided with KM graph. Also consider if y-axis should include full scale of probability 0.0 – 1.0 for accurate context.

Response 26:

Thank you for these suggestions. The number of trials appears in the risk tables at the bottom of each figure (figures 2,3, 4 and 5). We have changed the y-axis as suggested in all four survival curves figures (figures 2,3, 4 and 5). We have also made the changes in the titles by replacing “delay” by “time”.

*Comment 27:

Discussion - the discussion needs significant attention to better focus on and reflect the results from this analysis and their relevance to the colorectal cancer community

- The discussion (page 13, line 268) discusses ‘compliance’ however the objectives and methods of this paper do not support any inference about compliance and this should be deleted (i.e., analysis was not limited to phase 2-4 trials of FDA-regulated drug products and did not assess other appropriate reasons for results being available later than 12 months after completion)

Response 27:

We fully agree with the reviewer. As mentioned previously in comment 4, we have refocused our background and discussion on our main objectives. The background in our revised manuscript does not mention the FDAAA801 and we have deleted the part on compliance in the discussion section. Nevertheless, we have left a paragraph to discuss results posting on registries as follows:

Discussion section, pages 18 and 19:

“This is why submission of results on the registry independently of journal publication is crucial to improve transparency but still underfollowed as illustrated by our findings. Indeed, some registries require submission of results for selected trials within a defined time range after trial completion. Since 2007, the US Food and Drug Administration Amendments Act 801 (FDAAA 801) requires submission of trial results on ClinicalTrials.gov no later than one year after the primary completion date for applicable trials (see definition on https://www.clinicaltrials.gov/ct2/home). Similarly, for any interventional study that ended on or after 2014-07-21 and registered on the European registry (EudraCT), sponsors are requested to post trial results within 12 months following the end of the trial.”

“Also, regarding posting of results on ClinicalTrials.gov, we did not assess whether trials were considered as “applicable” according to the FDAAA 801; therefore, this result should be considered with caution as not all trials are required to post their results on the registry.”

“Many physicians from various fields, including oncologists, surgeons, pathologists, gastroenterologists, general practitioners and others, are involved in the management of CRC and should be aware of the waste of research in this field.”

*Comment 28:

- Page 14, line 274 specifies ‘enough time for publication’ however as noted earlier some studies had only 14 months; consider specifying as limitation

Response:

Thank you for this comment. We have added this comment as a limit in the discussion section of the revised manuscript:

Discussion section, page 19:

“Furthermore, considering the selected time range, some trials in our sample only had 14 months for assessment of results availability.”

*Comment 29:

- Page 14, line 275 mentions ‘the announcement’ – should specify who made this announcement; this discussion point is disjointed from the background that did not mention the ICMJE policy at all; recommend bringing the background and discussion into alignment

Response 29:

We agree with the reviewer. We have deleted that sentence in the discussion and moved it to the background section for more clarity. We have made the modification in the revised manuscript as follows:

Background section, page 3:

“Since the 2004 announcement by the International Committee of Medical Journal Editors of required registration for each new clinical trial before participant enrollment and as a condition for publication, there has been a strong increase of new trial registrations worldwide (1–3).”

*Comment 30:

- I expect there are additional references beyond reference 13 that could be sued to support the sentence ending on line 281 (page 14)

Response 30:

The reviewer is correct. We have updated the discussion section and added the following references:

- Turner EH, Matthews AM, Linardatos E, Tell RA, Rosenthal R. Selective publication of antidepressant trials and its influence on apparent efficacy. N Engl J Med. 2008 Jan 17;358(3):252–60

- Rising K, Bacchetti P, Bero L. Reporting bias in drug trials submitted to the Food and Drug Administration: review of publication and presentation. PLoS Med. 2008 Nov 25;5(11):e217; discussion e217

Discussion section, page 17:

“Previous works have shown that the results of many trials remain unpublished in various research fields (12–14). Similarly, despite public registration, about 50% of randomized controlled trials (RCTs) results are never published (15). One work published in 2013 had shown that results of nearly half of the oncology trials performed in the United States were not publicly available 3 years after completion (14).”

*Comment 31:

- Page 14, line 286 – specify what the ethical issue is

Response 31:

Thank you for this remark. We think it is unethical to not share results because there is always a risk for patients when participating in a new trial, especially when testing a new intervention (benefit/risk balance to assess). Also, not knowing if the tested intervention had a positive or negative effect on the initial condition is an issue. We have made the following changes in the revised version of the manuscript:

Discussion section, page 18:

“Also, not publishing results raises an ethical issue for patients who agree to participate in these trials (benefit/risk balance to assess when participating in a new trial).”

*Comment 32:

- Page 14, lines 291 -3. Unclear why IPD and ‘study documentations’ are being discussed as this was not a focus of this paper.

Response 32:

We fully agree with the reviewer and have deleted these elements in the discussion of the revised manuscript.

*Comment 33:

- Reference 21 is not appropriate for supporting the statement on page 14, lines 293-295 as it shows more results in ClinicalTrials.gov but can’t characterize as an ‘improvement’

Response 33:

The reviewer is correct. We have modified the sentence in the revised manuscript as follows:

Discussion section, page 17:

“Recent analysis performed on the ClinicalTrials.gov registry show an increase in the number of trials with posted results, but we cannot conclude on an improvement of trial results availability over time (22).”

*Comment 34:

Minor editorial

- Section headers of ‘search strategy for posted results’ and ‘search strategy for publication of results’ seem unnecessary, could be combined

- ‘overtime’ should be ‘over time’

Response 34:

Thank you, we have made the two requested changes in the revised version of the manuscript.

---

## [Decision Letter · Decision Letter 1]

17 Feb 2022

PONE-D-21-34103R1Availability of results of interventional studies assessing colorectal cancer over the past seven years.PLOS ONE

Dear Dr. Pellat,

Thank you for submitting your manuscript to PLOS ONE. After careful consideration, we feel that it has merit but does not fully meet PLOS ONE’s publication criteria as it currently stands. Therefore, we invite you to submit a revised version of the manuscript that addresses the points raised during the review process.

We look forward to receiving your revised manuscript.

Kind regards,

Tim Mathes

Academic Editor

PLOS ONE

Journal Requirements:

Reviewers' comments:

Reviewer's Responses to Questions

**Comments to the Author**

1. If the authors have adequately addressed your comments raised in a previous round of review and you feel that this manuscript is now acceptable for publication, you may indicate that here to bypass the “Comments to the Author” section, enter your conflict of interest statement in the “Confidential to Editor” section, and submit your "Accept" recommendation.

Reviewer #1: All comments have been addressed

Reviewer #2: All comments have been addressed

2. Is the manuscript technically sound, and do the data support the conclusions?

Reviewer #1: No

Reviewer #2: Yes

3. Has the statistical analysis been performed appropriately and rigorously? 

Reviewer #1: Yes

Reviewer #2: Yes

4. Have the authors made all data underlying the findings in their manuscript fully available?

Reviewer #1: Yes

Reviewer #2: Yes

5. Is the manuscript presented in an intelligible fashion and written in standard English?

Reviewer #1: Yes

Reviewer #2: Yes

6. Review Comments to the Author

Reviewer #1: (No Response)

Reviewer #2: The authors adequately addressed most of my prior comments. A few new (based on revisions) and remaining issues are noted here:

- Recommend revising the manuscript title to replace “over the past 7 years” to specify the actual years included in the analysis

- Lines 99 – 101 – it could be clarified that these are response options for the ‘primary purpose’ data element and the ‘intervention type’ data elements.

- Consider if the explanation of ‘open-access’ be instead included in the ‘eligibility of results’ section (i.e., did the reviewer as a third step note if the published results were open access?)

- Lines 158 – 159 – recommend specifying which data elements determine funding (was it sponsor and collaborators?)

- Table 2 – at a minimum denominator (# trials) needs to be added to the table column/row headers to provide appropriate context to the percentages, but could provide clarity by including numerator and denominator for each percentage

- Line 291 – appears to describe the results from Table 3 re: randomized trials but the data are different. Please verify (HR=1.2 (95%CI 0.9 – 1.6)) v. (HR=1.0 (95% CI 0.8 -1.4)).

- I don’t agree that the example of results before PCD demonstrates lack of currency or quality (Lines 358) but appreciate the authors qualification of a potential reason for this particular event

- Line 349 – misquotes the law and should be revised from "applicable trials" to “applicable clinical trials"

Note: I agree with the authors’ response (Response 6) to Reviewer #1's comment re: the inclusion of terminated studies in the analysis. Trials may be terminated for any number of reasons, including for interim findings of safety or efficacy, that make them important to include.

7. PLOS authors have the option to publish the peer review history of their article (what does this mean?). If published, this will include your full peer review and any attached files.

Reviewer #1: No

Reviewer #2: No

---

## [Author Response · Author response to Decision Letter 1]

18 Mar 2022

Reviewer #2 

The authors adequately addressed most of my prior comments. A few new (based on revisions) and remaining issues are noted here:

*Comment 1:

- Recommend revising the manuscript title to replace “over the past 7 years” to specify the actual years included in the analysis

Response 1:

Thank you, we have made the requested change in the new revised version of the manuscript.

Title: “Availability of results of interventional studies assessing colorectal cancer from 2013 to 2020.”

*Comment 2:

- Lines 99 – 101 – it could be clarified that these are response options for the ‘primary purpose’ data element and the ‘intervention type’ data elements.

Response 2:

Thank you for the suggestion. We have made the change as follows:

Modification in the methods section, line 99:

“We defined as management all trials with a primary purpose of screening, diagnosis, prevention (secondary and tertiary), treatment (including drugs, dietary supplement, behavioral intervention, device and surgery), and supportive care. Of note, these are response options proposed by ClinialTrials.gov for primary purpose and treatment/intervention types.”

*Comment 3:

- Consider if the explanation of ‘open-access’ be instead included in the ‘eligibility of results’ section (i.e., did the reviewer as a third step note if the published results were open access?)

Response 3:

We agree with the reviewer and have made the following changes in the manuscript:

Modification in the methods section, “eligibility criteria for results section”, line 150:

“We considered results to be available in open access when available on ClinicalTrials.gov and/or in an open access publication downloaded from our institution.”

*Comment 4:

- Lines 158 – 159 – recommend specifying which data elements determine funding (was it sponsor and collaborators?)

Response 4:

Thank you for this comment. Indeed, type of funding is determined by the “sponsor and collaborator” item from clinicaltrials.gov. We have made the following change in the manuscript, as well as in supplementary table 2:

Modification in the methods section, line 158:

“Type of funding (sponsors and collaborators) was divided in three categories”

*Comment 5:

- Table 2 – at a minimum denominator (# trials) needs to be added to the table column/row headers to provide appropriate context to the percentages, but could provide clarity by including numerator and denominator for each percentage

Response 5:

We agree with the reviewer and have added the denominators for each row in Table 2.

*Comment 6:

- Line 291 – appears to describe the results from Table 3 re: randomized trials but the data are different. Please verify (HR=1.2 (95%CI 0.9 – 1.6)) v. (HR=1.0 (95% CI 0.8 -1.4)).

Response 6:

Thank you for this comment. Table 3 only shows the results of the post-hoc multivariate analysis (after reviewing). In the text, we have left the results for the initial univariate analysis which does not appear in Table 3 (this explains the different HR values).

*Comment 7:

- I don’t agree that the example of results before PCD demonstrates lack of currency or quality (Lines 358) but appreciate the authors qualification of a potential reason for this particular event

Response 7:

Thank you for your understanding and for accepting this explanation in the discussion.

*Comment 8:

- Line 349 – misquotes the law and should be revised from "applicable trials" to “applicable clinical trials"

Response 8:

Thank you for this remark. We have made the correction in the revised version of the manuscri

---

## [Editor Report · Decision Letter 2]

22 Mar 2022

Availability of results of interventional studies assessing colorectal cancer from 2013 to 2020.

PONE-D-21-34103R2

Dear Dr. Pellat,

We’re pleased to inform you that your manuscript has been judged scientifically suitable for publication and will be formally accepted for publication once it meets all outstanding technical requirements.

Kind regards,

Tim Mathes

Academic Editor

PLOS ONE
---

## [Editor Report · Acceptance letter]

1 Apr 2022

PONE-D-21-34103R2 

Availability of results of interventional studies assessing colorectal cancer from 2013 to 2020. 

Dear Dr. Pellat:

I'm pleased to inform you that your manuscript has been deemed suitable for publication in PLOS ONE. Congratulations! Your manuscript is now with our production department. 

Kind regards, 

on behalf of

Dr. Tim Mathes 

Academic Editor

PLOS ONE